# Deciphering the Role of Copper Homeostasis in Atherosclerosis: From Molecular Mechanisms to Therapeutic Targets

**DOI:** 10.3390/ijms252111462

**Published:** 2024-10-25

**Authors:** Xuzhen Lv, Liyan Zhao, Yuting Song, Wen Chen, Qinhui Tuo

**Affiliations:** 1Key Laboratory for Quality Evaluation of Bulk Herbs of Hunan Province, School of Pharmacy, Hunan University of Chinese Medicine, Changsha 410208, China; 20232054@stu.hnucm.edu.cn; 2Key Laboratory of Vascular Biology and Translational Medicine, Medical School, Hunan University of Chinese Medicine, Changsha 410208, China; 14753872819@163.com (L.Z.); chenwen@biochen.org (W.C.); 3College of Integrative Chinese and Western Medicine, School of Medicine, Hunan University of Chinese Medicine, Changsha 410208, China; 19862120965@163.com

**Keywords:** copper, copper homeostasis, copper-induced cell death, atherosclerosis

## Abstract

Cardiovascular disease (CVD) is a leading cause of death globally, with atherosclerosis (AS) playing a central role in its pathogenesis as a chronic inflammatory condition. Copper, an essential trace element in the human body, participates in various biological processes and plays a significant role in the cardiovascular system. Maintaining normal copper homeostasis is crucial for cardiovascular health, and dysregulation of copper balance is closely associated with the development of CVD. When copper homeostasis is disrupted, it can induce cell death, which has been proposed to be a novel form of “cuproptosis”, distinct from traditional programmed cell death. This new form of cell death is closely linked to the occurrence and progression of AS. This article elaborately describes the physiological mechanisms of copper homeostasis and explores its interactions with signaling pathways related to AS. Additionally, we focus on the process and mechanism of cell death induced by imbalances in copper homeostasis and summarize the relationship between copper homeostasis-related genes and AS. We also emphasize potential therapeutic approaches, such as copper balance regulators and nanotechnology interventions, to adjust copper levels in the body, providing new ideas and strategies for the prevention and treatment of CVD.

## 1. Introduction

According to the World Heart Federation’s “World Heart Report 2023”, CVD remains the foremost cause of mortality globally, claiming the lives of 20.5 million individuals annually and representing nearly a third of all worldwide deaths [1]. Atherosclerosis (AS), the fundamental pathological process underlying CVD, manifests as a chronic inflammatory condition typified by the accumulation of lipids, cholesterol, and other substances within the arterial walls, leading to the formation of plaques. The trace element copper, indispensable for human health, assumes a pivotal role in an array of biological processes and significantly influences cardiovascular function [2,3]. Serving as an integral component of numerous enzymes and proteins, notably within the active sites of enzymes such as superoxide dismutase [4,5], copper facilitates the neutralization of free radicals in the body, thereby shielding blood vessels against oxidative stress-induced damage. Consequently, copper’s involvement is intrinsically associated with various signaling pathways and the maintenance of cardiovascular well-being.

In recent years, the significance of copper homeostasis in cardiovascular health has garnered extensive attention. Copper homeostasis represents the equilibrium state of copper concentrations within the body and encompasses intricate processes such as absorption, transport, storage, and excretion of copper. In the context of cardiovascular disease (CVD), perturbations in copper homeostasis can exacerbate disease progression, revealing its “double-edged sword” characteristic: copper deficiency compromises vascular elasticity and heightens the risk of arteriosclerosis [6,7,8,9], while copper overload catalyzes oxidative stress and inflammatory cascades, thereby hastening the advancement of atherosclerosis [10,11,12]. Notably, supraphysiological levels of copper may induce an unprecedented form of cellular demise termed “cuproptosis”, which diverges from conventional programmed cell death modalities, such as apoptosis, necrosis, and autophagy [13], offering a novel perspective on the implications of copper in CVD.

In this study, original literature was searched and collected from databases such as PubMed, Web of Science, and Google Scholar. The following search keywords were used: copper; copper homeostasis; copper-induced cell death; Atherosclerosis; and their combinations. There was no limitation on publication year, with the last search conducted on 24 July 2024. A total of 1624 articles were retrieved. After excluding duplicate articles, review articles, meta-analyses, standalone abstracts, and articles irrelevant to the research topic, 129 articles were included. Furthermore, this manuscript succinctly recapitulates select advancements in the realm of copper and cardiovascular wellness (Figure 1). This article endeavors to scrutinize copper homeostasis, copper-induced cytotoxicity, and their correlation with atherosclerosis, with the aspiration of guiding future investigative trajectories in related disciplines [14,15,16,17,18,19,20,21,22].

## 2. Physiological Mechanisms of Copper Homeostasis

### 2.1. Copper Biochemistry: Metabolism and Transport Mechanisms

Copper absorption starts when divalent ions Cu(II) are ingested through the food chain into the human body [23]. Subsequently, these ions are efficiently absorbed in the stomach, duodenum, and epithelial cells of the small intestine. During this process, Cu(II) encounters metal reductases from the STEAP family and duodenal cytochrome b (Dcytb), which reduce it to monovalent ions Cu(I). Subsequently, Cu(I) enters the cell through the actions of copper transporter 1 (CTR1, also known as SLC31A1), copper transporter 2 (CTR2, also known as SLC31A2), and divalent metal transporter 1 (DMT1) [24].

From there, these copper ions are transported to the liver via the portal vein, primarily relying on the transport functions of ATP7A and ATP7B [25]. Upon reaching the liver, copper begins to play its core role in the metabolism process, responsible for the storage, utilization, and excretion of copper [26]. To counteract the potential toxicity of copper, organisms adopt various protective measures. Firstly, when there is an excess of Cu(I) in the body, these ions bind with glutathione (GSH) and metallothionein (MT) to form a complex called a chelate, which allows for temporary copper storage and prevents its free ions from damaging cells [27]. Additionally, the copper ion pool in mitochondria plays a similar role, temporarily accumulating copper ions to regulate their concentration and protect mitochondria from copper toxicity.

In the process of copper ion handling and storage, the transport proteins ATP7A and ATP7B in the Golgi apparatus also play a key role. These two proteins can transport Cu(I) into the trans-Golgi network (TGN), which not only aids in temporary copper storage but is also crucial for maintaining intracellular copper ion balance and stability [28]. Subsequently, copper ions are transported to various organs (kidneys, brain, heart, etc.) for utilization. In the transport vessels, copper ions exist in the form of complexes with various ligands, forming copper complexes with amino acids such as histidine, cysteine, and aspartic acid [29], and also forming looser complexes with proteins such as albumin (Alb), amino acids (AA), macroglobulin, ceruloplasmin (Cp), MT, and human serum albumin (HSA) [30].

These complexes effectively promote copper transport in the blood and prevent excessive accumulation in the body. Finally, excess copper ions are excreted from the body, mainly through bile. Among them, ATP7A and ATP7B are key excretory proteins that can transport copper ions from hepatocytes into bile, thereby promoting copper excretion (Figure 2).

### 2.2. Copper Proteins in Copper Homeostasis

Copper homeostasis is a complex physiological process involving the absorption, distribution, utilization, and excretion of copper. Various copper proteins play a key role in ensuring the balance of copper ions within the body. These include copper complexes, copper chaperones, and copper transporters [31]. Copper–amino acid complexes (such as histidine–copper and cysteine–copper complexes) are involved in the absorption process of copper, increasing its solubility and bioavailability and promoting the entry of copper from the intestine into cells.

Once inside the cell, copper ions are transferred to specific copper complexes. These include Cu,Zn-superoxide dismutase (SOD1) [32], which uses copper ions for catalytic reactions and protects cells from oxidative stress damage; cytochrome c oxidase (CCO) [33], a key component of the mitochondrial respiratory chain that uses copper ions to catalyze its activity and participate in energy metabolism and ATP synthesis; ceruloplasmin (CP) and metallothionein (MT), which transport and store copper ions and protect cells from oxidative damage; and glutathione, which relies on intracellular copper ions to exert antioxidant effects.

The transfer of copper ions requires the participation of multiple copper transporters and chaperones. CTR1 and CTR2 are the main channels for copper entry into cells. They are located on the cell membrane and can transport copper from the external environment or intestinal sites into the cell. In humans, the transport and utilization of copper are precisely regulated by various proteins. CTR1 is highly expressed in the liver and small intestine, and its key function is to transport copper (I) from the extracellular environment across the membrane into the cytoplasm [34]. Meanwhile, CTR2 is located on the vacuolar membrane of liver cells, and its main function is to release stored copper (I) from vacuoles into the cytoplasm [35].

Within the cell, the transfer of copper ions is handled by SLC25A3, which transports cytoplasmic copper ions to mitochondria [36], a crucial step for the assembly of CCO. SLC25A3 is located on both sides of the mitochondrial inner and outer membranes, ensuring effective transfer of copper ions [37]. Additionally, the assembly and function of CCO also depend on other auxiliary proteins. Cytochrome C oxidase copper chaperone 17 (COX17) is responsible for transporting copper ions from the cytoplasm to the mitochondrial matrix and receiving copper ions transferred by SLC25A3. Cytochrome C oxidase copper chaperone 11 (COX11) also participates in the assembly process of CCO. In mitochondria, SCO1 and SCO2 capture copper ions from the copper ion pool and further transport them to the assembly site of CCO. Research has shown that SCO1 and SCO2 may interact with mitochondrial matrix copper chaperone proteins such as COX17, directly participating in the assembly of CCO. Mitochondrially encoded cytochrome c oxidase I (MTCO1, also known as COX1) and mitochondrially encoded cytochrome c oxidase II (MTCO2, also known as COX2), as important subunits of CCO, contain binding sites for copper ions, which are essential for the catalytic activity of CCO [38].

To maintain copper homeostasis within cells, human antioxidant protein 1 (Atox1) is responsible for transferring copper from CTR1 to the copper pumps ATP7A and ATP7B located in the trans-Golgi network [39]. Recent technological developments, such as engineered ascorbate peroxidase (APEX2) proximity labeling technology, have enabled researchers to capture and analyze transient and weak interactions between ATOX1 and its downstream copper proteins. Studies have shown that Atox1 can transfer copper ions to the nucleus and further pass them to cysteine-rich protein 2 (CRIP2), causing a change in the secondary structure of CRIP2, thereby promoting its ubiquitination and degradation. Therefore, copper proteins play an important regulatory role in copper homeostasis and copper transport, ensuring that copper ions are utilized and regulated appropriately to maintain normal operation of the mitochondrial respiratory chain and cellular energy balance. Once copper homeostasis is disrupted, it can lead to metabolic disruption and even cell death.

## 3. Copper Homeostasis and Atherosclerosis Signaling Pathways

Copper plays a crucial role in the human body, being an integral component of many key enzymes. It is closely associated with endothelial dysfunction, macrophage-driven inflammation, proliferation and migration of smooth muscle cells, and dysregulation of lipid metabolism (Figure 3). These factors can directly or indirectly promote the development and further exacerbation of atherosclerosis.

Low-density lipoprotein (LDL) penetrates the endothelial barrier and enters the inner wall of the artery, where it is transformed into oxidized low-density lipoprotein (Ox-LDL) through copper-catalyzed oxidation [40]. This process not only harms endothelial cells but also stimulates the growth and movement of smooth muscle and endothelial cells. Concurrently, it stimulates the release of chemotactic factors, prompting monocytes to transform into macrophages that engulf Ox-LDL, leading to the formation of foam cells [41]. Smooth muscle cells in the artery migrate to the intima and transform into foam cells, participating in the formation of the fibrous cap and the necrotic core of atherosclerotic plaques, ultimately resulting in the development of atherosclerotic plaques.

### 3.1. Copper Proteins in Vascular Endothelial Cells

Disturbances in copper homeostasis can cause multifaceted harm to the cardiovascular system. An excess of copper leads to elevated intracellular copper ion concentrations, promoting the formation of free radicals, especially reactive oxygen species (ROS) [42,43,44]. These free radicals directly damage cellular components, including lipids, proteins, and DNA, thereby inducing oxidative stress. Within the vascular system, such oxidative stress causes endothelial dysfunction and promotes the formation of atherosclerotic plaques [45]. Specifically, copper can enhance the production of ROS in endothelial cells, activating nuclear factor kappa-B (NF-κB), a key transcription factor involved in regulating the expression of various genes related to inflammation and immune responses. Under the influence of copper, the activation of NF-κB leads to changes in a series of adhesion molecules. The expression of adhesion molecules such as vascular cell adhesion molecule-1 (VCAM-1), intercellular adhesion molecule-1 (ICAM-1), and E-selectin increases. The enhanced expression of these adhesion molecules on the surface of endothelial cells facilitates the interaction between white blood cells and endothelial cells, thereby promoting the occurrence and development of inflammatory responses [46,47]. Moreover, inflammatory cytokines trigger a series of complex molecular events by promoting elevated intracellular copper levels. This increase in copper levels leads to the activation of X-linked inhibitor of apoptosis (XIAP) [48], which in turn promotes the activation of NF-κB. NF-κB, as a key inflammatory regulator, further exacerbates the inflammatory response in atherosclerotic endothelial cells, driving disease progression. XIAP not only plays a role in inhibiting inflammation but also has a significant function in regulating apoptosis. However, when XIAP binds to copper, it undergoes ubiquitination and proteasomal degradation, a process that causes XIAP to lose its ability to inhibit apoptosis [49], affecting the balance between cell survival and death, and consequently, the stability of plaques and the severity of the disease.

Meanwhile, copper also affects the Kelch-like ECH-associated protein 1 and nuclear factor erythroid 2-related factor 2 (KEAP1-NRF2) complex, which dissociates under oxidative stress, allowing NRF2 to enter the nucleus and activate a series of antioxidant response elements, such as heme oxygenase-1 (HO-1), to mitigate oxidative damage to cells [50,51]. Copper is also an essential cofactor for hypoxia-inducible factor-1α (HIF-1α) and, through the mediation of copper chaperone protein (CCS), it participates in maintaining the stability and regulating the activity of HIF-1α [52,53]. A copper deficiency hinders its activity, while HIF-1α stabilizes and activates under hypoxic conditions, regulating the expression of genes related to hypoxia adaptation, such as vascular endothelial growth factor α (VEGFα), NOS3, etc. These genes are closely associated with processes like angiogenesis, vasoconstriction, and antioxidative stress [54].

### 3.2. Copper in Macrophages

When present in excess, copper ions act as a catalyst for triggering inflammatory responses, thereby exacerbating the inflammation of blood vessel walls and accelerating the progression of CVD [55]. Specifically, copper can stimulate the release of chemotactic factors, which serve as guiding signals, leading monocytes to transform into macrophages and prompting them to engulf LDL. Subsequently, these macrophages further transform into foam cells, a process that is a significant hallmark of early atherosclerotic lesions. Additionally, the activation of the mitogen-activated protein kinase (MAPK) signaling pathway requires the participation of copper ions. Catalyzed by copper, it can bind to mitogen-activated protein kinase kinase 4 (MEK4), thereby promoting the activation of downstream c-Jun N-terminal kinase (JNK) and P38 [56]. These molecules are key factors in regulating inflammatory responses, and under the induction of copper, they further intensify the inflammatory reaction. Meanwhile, copper-induced oxidized low-density lipoproteins can also induce oxidative stress within cells, and in this process, activate Janus Kinase 2 (JAK2), which further activates signal transducer and activator of transcription (STAT) to regulate cellular inflammation [57,58,59].

### 3.3. Copper in Vascular Smooth Muscle Cells

The mechanism of action of copper in smooth muscle cells plays a crucial role in receptor tyrosine kinase-associated signaling pathways. Copper regulates the phosphorylation status of receptor tyrosine kinases (RTKs) or interacts directly with RTKs, thereby influencing RTK activity [60,61]. This process enhances the activation of an important member of the RTK family—the platelet-derived growth factor receptor (PDGFR), subsequently triggering the phosphorylation of downstream extracellular regulated protein kinases (ERK) and proline-rich tyrosine kinase (PYK2), ultimately leading to the proliferation and migration of smooth muscle cells [62]. On the other hand, copper ions are also believed to activate downstream molecules by acting on the phosphoinositide 3-kinase (PI3K-AKT) signaling pathway [63]. The activation of AKT further catalyzes the phosphorylation and subcellular redistribution of forkhead box O1a (FoxO1a) and forkhead box O3 (FoxO3), regulating the proliferation of vascular smooth muscle cells (VSMCs) through the Akt-FoxO-PCK2 pathway [64]. Additionally, copper can directly bind to mitogen-activated protein kinase kinase 1 (MEK1), promoting the phosphorylation of extracellular signal-regulated kinase 1/2 (ERK1/2) [65], thereby further regulating the growth of smooth muscle cells [66]. Autophagy is an intracellular degradation process that maintains cellular homeostasis by removing damaged organelles and proteins. In smooth muscle cells, autophagy helps remove damaged organelles and abnormal proteins, thereby preventing cellular dysfunction [67]. Copper can directly or through the influence of the mTOR target protein bind to Unc-51 like autophagy activating kinase (ULK), promoting the phosphorylation and activation of autophagy related 13 (ATG13), leading to the formation of the autophagy complex [68], ultimately affecting the survival, migration, proliferation, and death of smooth muscle cells.

### 3.4. Copper in Metabolism

Copper plays a crucial role in cellular metabolism, particularly in lipid and glucose metabolic pathways. In a rabbit model of atherosclerosis, the expression of pyruvate kinase isoform M2 (PKM2) in VSMCs was observed to be more abundant in the intima than in the media [69]. Treatment with a specific inhibitor of PKM2 significantly slowed the progression of atherosclerosis. PKM2, a key rate-limiting enzyme in the glycolytic pathway, has been shown to be influenced by copper [69]. This finding suggests that targeting the PKM2-dependent glycolytic process could be a new strategy for treating atherosclerosis. Recent studies have shown that copper can reduce the expression of S6 Kinase 1 (S6K1) and its downstream glycolysis-related molecules glucose transporter type 1 (GLUT1), PKM2, and lactate dehydrogenase A (LDHA) [70], indicating that copper may delay the development of AS by intervening in the glycolytic pathway. Additionally, copper can regulate lipolysis by interacting with the cysteine residues of phosphodiesterase 3B (PDE3B) [71]. Research reveals that copper-induced oxidative stress can promote the accumulation of Nrf2 on the peroxisome proliferator-activated receptor gamma (PPARγ) promoter, thereby activating target gene transcription and further promoting adipogenesis [72]. Meanwhile, copper ions enhance the activity of the synthesis of cytochrome c oxidase 1-liver kinase B1-AMP-activated protein kinase (SCO1-LKB1-AMPK) complex, increase the expression of peroxisome proliferator-activated receptor gamma coactivator 1 alpha (PGC1α), and, subsequently, stimulate the expression of fatty acid oxidation (FAO) genes and mitochondrial biosynthesis genes, promoting mitochondrial biogenesis and fatty acid oxidation, thereby effectively enhancing lipolysis [73]. These findings indicate that copper can influence the development of AS by intervening in glycolysis and lipid metabolism.

## 4. Copper Homeostasis Imbalance-Induced Cell Death

Research on copper-induced cell death dates back to the 1980s, yet its exact mechanism remains elusive. Scientists have observed that in environments with high concentrations of copper ions, cells undergo a series of morphological changes, including cytoplasmic shrinkage, nuclear condensation, DNA fragmentation in the nucleus, and vesicular changes in the cell membrane. Studies by Pan M et al. have shown that prolonged exposure to copper leads to incompleteness in the double-membrane structure of mitochondria in mice [74]. Wang et al. found that copper exposure results in enlarged liver cells in mice, increased lipid droplets and glycogen granules in the cytoplasm, and swelling and vacuolization of mitochondria and endoplasmic reticulum [75]. Seo Y’s research confirmed that high concentrations of copper ions damage endothelial cells, causing changes in cell morphology and structure, and disrupting the integrity of the cell membrane [19]. Singh et al. discovered that copper causes DNA fragmentation and damages vesicular structures and cristae in human peripheral blood mononuclear cells (PBMCs) [76].

Excessive levels of copper, especially through the promotion of the Fenton reaction, significantly increase the production of ROS and free radicals, impacting cell structure and function. These free radicals trigger lipid peroxidation, protein damage, and DNA damage, leading to the destruction of cell structure and function [77,78,79]. In the Fenton reaction, copper efficiently catalyzes the generation of ROS through the alternating conversion of its two main oxidation states, Cu^+^ and Cu^2+^. These ROS rapidly consume intracellular antioxidant molecules like GSH to neutralize these reactive oxygen species, thereby protecting the cell from oxidative damage. However, when the consumption of GSH exceeds its synthesis capability, the cell faces severe oxidative stress [80]. Additionally, excess copper encourages GSH coupling reactions, increasing cytotoxicity and accelerating the process of cell death. In this process, GSH, as an important antioxidant molecule, plays a crucial role in maintaining cellular redox balance [81].

When copper ions enter the cell and catalyze the production of large amounts of ROS, GSH is rapidly consumed to neutralize these reactive oxygen species. Meanwhile, ROS attack unsaturated fatty acids on the cell membrane, triggering lipid peroxidation reactions and forming toxic by-products such as malondialdehyde (MDA) and 4-hydroxynonenal (4-HNE). These by-products cause significant damage to cell structure and DNA. As a result, the integrity of the cell membrane is compromised, leading to impaired function, altered fluidity, and increased permeability, all of which can ultimately lead to cell rupture [82,83]. ROS can directly oxidize protein side chains, causing peptide chain breakage, cross-linking, and changes in tertiary structure, affecting enzyme activity and protein function. The stability of p53 protein is reduced, activating Bax and inhibiting Bcl-2, leading to apoptosis [84].

In the nucleus, ROS such as OH· can directly react with DNA molecules, causing base modification, sugar group destruction, and chain breakage, activating the p53 gene. This activation triggers genes such as damage-regulated autophagy modulator (DRAM) and sestrin2 (Sesn2), leading to more autophagosomes [85,86]. Meanwhile, copper ions induce oxidative stress that damages mitochondrial membranes, leading to a decrease in membrane potential, which in turn affects ATP synthesis and cellular energy supply, activating AMPK. Then AMPK inhibits mTORC1, activates unc-51 like autophagy activating kinase 1 (ULK1), and promotes the formation of autophagolysosomes. Excessive autophagy ultimately leads to cell death [87,88].

In 2022, Tsvetkov et al. proposed a novel type of cell death induced by copper, namely cuproptosis [13]. This mode of death differs from traditional apoptosis or necrosis; it involves copper binding to mitochondrial lipoylated proteins, causing their abnormal aggregation and interfering with the tricarboxylic acid cycle and aerobic respiration, ultimately leading the cell into a toxic stress state and death. This unique pathway of cell death is caused by excessive copper in the cell, different from typical oxidative stress-induced cell death. Excess copper in mitochondria disrupts energy metabolism, releases cytochrome c, and activates caspase-9, thus triggering apoptosis [89,90]. Moreover, copper can bind to lipoylated proteins, inhibiting mitochondrial metabolic functions, leading to the loss of lipoylated proteins and mitochondrial respiratory iron–sulfur cluster proteins, inducing protein toxic stress and cell death. Protein thiolation in the tricarboxylic acid (TCA) cycle is a key feature of cuproptosis [91]. Copper also inhibits the ubiquitin–proteasome system (UPS), reducing cellular protease activity, blocking cell growth, and leading to cell death [92]. Furthermore, disulfiram–Cu^2+^ complexes target the upstream signaling pathways of the proteasome system, hindering ubiquitin-dependent ATP synthase, leading to the accumulation of ubiquitinated proteins and ultimately resulting in cell death [93] (Figure 4).

## 5. Copper Homeostasis-Related Genes and AS: Potential Value

Since the inception of research, there has been a growing interest in copper homeostasis, particularly with the introduction of the concept of copper-induced cell death. Most researchers have begun to delve into the correlation between copper homeostasis and AS. In the treatment research of AS, the regulation of copper homeostasis and the potential impact of its related genes have become an emerging and noteworthy field (Table 1). Previous studies have confirmed that an increase in serum copper ion concentration is closely linked to human CVD. At the same time, high concentrations of copper ions have been detected in human carotid plaque tissue [94], further highlighting the significant role of copper ions in the progression of AS.

Tsvetkov and colleagues successfully identified 10 genes associated with altered risk of copper-induced cell death through the use of whole-genome knockout technology, and further observed the impact of these genes on cellular phenotypes. In this study, they discovered seven genes that positively regulate copper-induced cell death and three genes that negatively regulate it. The positive regulators include ferredoxin 1 (FDX1), lipoyl synthase (LIAS), lipoyltransferase 1 (LIPT1), dihydrolipoyl dehydrogenase (DLD), dihydrolipoamide S-acetyltransferase (DLAT), pyruvate dehydrogenase E1 component subunit alpha (PDHA1), and pyruvate dehydrogenase beta subunit (PDHB), which play a promoting role in the process of copper-induced cell death. The negative regulators are metal response element binding transcription factor 1 (MTF1), glutaminase (GLS), and CDKN2A, which inhibit the process of copper-induced cell death [95]. Through the analysis of copper homeostasis, researchers found that copper kinases ceruloplasmin (CP), superoxide dismutase (SOD), lysyl oxidase (LOX), copper transporters SLC31A1, SLC31A2, ATP7A, ATP7B, and copper chaperone proteins ATOX1, CCS, COX11, and COX17 can also inhibit copper-induced cell death by regulating copper ion concentration. These genes or proteins play a central role in the pathological mechanism of copper-induced cell death, and their functions and descriptions are listed in detail in Table 1. It is worth noting that these genes perform key functions within organisms and are often closely related to energy metabolism and angiogenesis processes.

With the introduction of the concept of cuproptosis, an increasing number of scholars have begun to focus on the role of these genes in AS. Research has shown that the expression levels and clinical significance of these genes in AS hold significant research value. Significant differences in the expression of genes such as LIAS, LIPT1, DLAT, PDHB, MTF1, and GLS were observed between patients with stable coronary artery disease and those with acute myocardial infarction (AMI) [96,97]. Similarly, Chen et al. observed increased expression of genes such as SLC31A1, SLC31A2, and FDX1 in atherosclerotic plaques, while the expression of SOD1 and GLS decreased, indicating that copper metabolism plays different roles in various stages of AS. In early-stage atherosclerosis, genes such as ATP7A, GLS, DLD, ATP7B, and PDHA1 are highly expressed, whereas in later stages, the expression levels of FDX1, CDKN2A, SLC31A1, and MTF1 increase [98]. These changes in copper metabolism and related gene expression may affect the development of AS through mechanisms such as abnormal vascular development, mitochondrial dysfunction, and various stress responses [99], ultimately promoting the disease.

**Table 1 ijms-25-11462-t001:** Potential value of copper homeostasis-related genes in the treatment of AS.

Gene	Description	Aliases	Distribution	Subcellular Locations	Role in Copper Homeostasis	Clinical Values	Ref.
SLC31A1	Solute Carrier Family 31 Member 1	COPT1, CTR1, NSCT	Liver, small intestine	Cell membrane, early endosomes, and recycling endosomes	High-affinity copper transporter mediating the entry of copper ions from the extracellular environment into cells	Emerging diagnostic biomarker and therapeutic target for acute myocardial infarction and arteriosclerosis	[22,25,99]
SLC31A2	Solute Carrier Family 31 Member 2	COPT2, CTR2, hCTR2	Esophagus mucosa, duodenum, gall bladder, adipose, liver	Late endosomes and lysosomes inner membranes	Copper ion transmembrane transporter that exports copper (I) from the lumen of the vesicle to the cytoplasm, participating in copper isolation and regulation	Highly expressed in arteriosclerotic plaques	[25,98]
ATP7A	ATPase copper transporting alpha	MK; MNK; HMNX; DSMAX; SMAX3	Small intestine, lung, kidney, muscle	Golgi or plasma membrane	Copper transporter that moves cytoplasmic copper ions into the lumen of the Golgi apparatus and further distributes them to various copper-dependent enzymes and proteins.	Potential therapeutic target for inflammatory vascular diseases	[25,30,64,100,101]
ATP7B	ATPase copper transporting beta	WD; PWD; WC1; WND	Liver, small intestine, kidney, placenta	Liver, small intestine, kidney, placenta	ATP7B is responsible for transporting copper from the cytoplasm to the Golgi apparatus for use by copper-dependent enzymes and proteins.	--	[25,99]
SLC25A3	solute carrier family 25 member 3	PHC; PTP; PiC; OK/SW-cl.48	Mitochondria	Mitochondrial outer membrane	Copper transporter responsible for importing copper into the mitochondria	--	[35,36]
COX11	cytochrome c oxidase copper chaperone COX11	COX11P; MC4DN23	Liver	Mitochondrial membrane	Copper chaperone protein that acts as an assembly factor for COX, not directly involved in copper transport or metabolism.	--	[37]
COX17	cytochrome c oxidase copper chaperone COX17	--	Heart, muscle	Cytosol	Copper transport protein responsible for transporting copper from the cytoplasm to the mitochondria. This process is crucial for maintaining the function of cytochrome c oxidase (COX), a copper-containing enzyme whose activity depends on the presence of copper ions.	--	[37]
ATOX1	antioxidant 1 copper chaperone	ATX1; HAH1	Liver	Cytoplasm	Copper chaperone protein that binds and transports copper ions from the cytoplasm to the ATPase proteins located in the trans-Golgi network.	Potential therapeutic target for VSMC migration and inflammation-related vascular diseases.	[39,102,103,104]
CCS	copper chaperone for superoxide dismutase	--	Liver, spleen	Cytoplasm	Copper chaperone for superoxide dismutase (CCS) that delivers copper ions to the copper/zinc superoxide dismutase (SOD1) protein.	--	[51,52]
CP	ceruloplasmin	CP-2; AB073614	Liver	Plasma (extracellular space)	A copper-containing metalloprotein, one of the main transporters of copper in the body, facilitates the transport and distribution of copper, ensuring its efficient delivery to various cells and tissues that require it, while preventing the toxic effects of free copper ions.	A potential biomarker for the risk of arteriosclerotic thrombosis	[105]
SOD	superoxide dismutase 1	ALS; SOD; ALS1; IPOA; STAHP; hSod1; HEL-S-44; homodimer	Liver, kidney, heart	Cytosol, mitochondria, extracellular matrix	A copper-containing metalloenzyme that catalyzes the dismutation of superoxide anion radicals into oxygen and hydrogen peroxide, thus protecting cells from damage by reactive oxygen species.	Improves lipid and glucose levels in atherosclerotic mice.	[106,107]
LOX	lysyl oxidase	AAT10	Fat, gallbladder	Extracellular matrix	-	A copper-dependent amine oxidase essential for the cross-linking of collagen and elastin, a process necessary for maintaining the structure and function of connective tissue.	[108,109,110,111,112,113,114]
FDX1	ferredoxin 1	ADX; FDX; LOH11CR1D	Liver and kidney	Mitochondria	Encoding iron–sulfur proteins, it is crucial for various metabolic reactions within the mitochondria, including the synthesis and metabolism of steroids, vitamin D, and bile acids. It acts as a key regulator in copper ion carrier-induced cell death and is an upstream regulator of cellular protein lipidation, participating in the synthesis of the electron transport chain and iron–sulfur clusters.	Highly expressed in atherosclerotic plaques and also highly expressed in a newly discovered subset of macrophages.	[95,115]
LIAS	lipoic acid synthetase	LS; LAS; LIP1; PDHLD; HGCLAS; HUSSY-01	Liver	Mitochondria	The encoded protein belongs to the biotin and lipoic acid synthetase family and is a crucial iron–sulfur enzyme that catalyzes the final step in the biosynthesis pathway of lipoic acid. It directly binds to FDX1, enhancing its role in cellular protein lipidation.	Overexpression of LIAS significantly reduces the size of atherosclerotic lesions in the aortic sinus.	[116,117,118]
LIPT1	lipoyltransferase 1	LIPT1D	Liver	Mitochondria	LIPT1 is responsible for the transfer of lipoic acid to proteins. This process is regulated by FDX1 and involves the lipoylation of DLAT.	Regulates mitochondrial function and lipid metabolism processes	[95]
DLD	dihydrolipoamide dehydrogenase	E3; LAD; DLDD; DLDH; GCSL; PHE3; OGDC-E3	Expressed at higher levels in the liver, heart, and kidneys	Mitochondria	The E3 subunit of the pyruvate dehydrogenase complex.	Associated with energy metabolism and lipid metabolism	[119,120]
DLAT	dihydrolipoamide S-acetyltransferase	E2; PBC; DLTA; PDCE2; PDC-E2	Expressed at higher levels in the liver.	Mitochondria	Excess copper leads to the aggregation of dihydrolipoamide S-acetyltransferase (DLAT), which is associated with the mitochondrial tricarboxylic acid (TCA) cycle, resulting in proteotoxic stress.	Ischemic cardiomyopathy immune biomarkers.	[121]
PDHA1	pyruvate dehydrogenase E1 subunit alpha 1	PDHA; PDHAD; PHE1A; E1alpha; PDHCE1A	Heart, fat, liver	Mitochondria	A component of the pyruvate dehydrogenase complex E1 subunit, a gene regulating cuproptosis.	A crucial link between glycolysis and the tricarboxylic acid cycle, it also directly impacts fatty acid metabolism and energy production.	[98,122]
PDHB	pyruvate dehydrogenase E1 subunit beta	PDHBD; PHE1B; E1beta; PDHE1B; PDHE1-B	Highly expressed in the heart and liver	Mitochondria	A component of the pyruvate dehydrogenase complex E1 subunit, a gene regulating cuproptosis.	-	[95,96,97]
MTF1	metal regulatory transcription factor 1	ZRF; MTF-1	Highly expressed in the liver	nucleus	Encoding a crucial transcription factor, it regulates the expression of genes related to heavy metal stress. In maintaining copper homeostasis, MTF1 senses changes in intracellular copper ion concentration, thereby controlling the expression of a series of copper-binding proteins and copper transport proteins to maintain the dynamic balance of copper ions within the cell.	Modulating lipid metabolism to reduce the risk of atherosclerosis caused by a high-fat diet.	[97,123]
GLS	glutaminase	GAC; GAM; KGA; GLS1; AAD20; DEE71; GDPAG; CASGID; EIEE71	Widely expressed in the liver and small intestine.	Mitochondria	GLS, as a key enzyme in glutamine metabolism, is influenced by intracellular copper ion concentrations.	Downregulation of GLS expression in atherosclerotic plaques.	[124]
CDKN2A	cyclin dependent kinase inhibitor 2A	ARF; MLM; P14; P16; P19; CAI2; CMM2; INK4;	Expressed at higher levels in the heart and muscle tissues	nucleus	Genes regulating cuproptosis	Plays a key role in cellular aging, and in atherosclerosis models, the mRNA levels of CDKN2A are reduced.	[125,126,127,128]

The copper transporter protein family is crucial for the regulation and transport of intracellular copper ions, with its members not only affecting angiogenesis and inflammatory responses but also being associated with the occurrence and development of various cardiovascular diseases. In particular, the important family members SLC31A1 and SLC31A2 have significantly higher expression in atherosclerotic plaques than in normal vascular cells, with particularly notable expression in macrophages, suggesting their potential roles in the formation process of AS [22].

The protein CTR1, encoded by SLC31A1, is a transmembrane protein responsible for regulating the intake of intracellular copper ions and maintaining copper ion homeostasis. Research has shown that CTR1 can promote platelet-derived growth factor (PDGF)-induced migration of VSMCs, potentially playing a key role in the development of atherosclerotic lesions [59]. When CTR1 is depleted, it affects VEGFR-2 signal transduction in endothelial cells, thereby affecting angiogenesis [129]. Disrupting copper homeostasis by knocking out CTR1 or ATOX1 proteins can inhibit angiogenesis in endothelial cells [130,131,132].

CTR2 is also a transmembrane protein capable of transporting copper across the cell membrane and increasing cytoplasmic copper levels, with its expression regulated by Cu status. In atherosclerotic plaques, the expression of SLC31A2 is higher than in normal tissues, showing potential as a diagnostic biomarker for AS [98].

During the development of AS, vascular inflammation plays a key role. ATP7A has anti-inflammatory effects in VSMCs, helping to slow the progression of AS [100,101]. microRNA-125b can directly target and inhibit the expression of ATP7A, thereby affecting the development of vascular inflammation [20]. ATP7A participates in the regulation of VSMCs migration through interaction with IQ motif-containing GTPase-activating protein 1 (IQGAP1) [20].

In ApoE^−/−^ mice on a high-fat diet and infused with angiotensin II, ATOX1 is present in the nuclei of inflamed aortas [102]. Further research has shown that a novel interaction between ATOX1 and tumor necrosis factor receptor-associated factor 4 (TRAF4) in endothelial cells is involved in the regulation of inflammatory responses. This copper-dependent binding of ATOX1 with TRAF4 is crucial for ATOX1 nuclear translocation and ROS-dependent inflammatory response in TNF-α stimulated endothelial cells [103]. Therefore, the ATOX1-TRAF4 axis may serve as a potential therapeutic target for vascular inflammatory diseases such as AS.

Additionally, the migration of VSMCs is crucial for neointimal formation after vascular injury and atherosclerotic lesion formation. Atox1 is involved in copper-induced cell growth and is upregulated in damaged vessels, colocalizing with copper transport protein ATP7A. Atox1 participates in neointimal formation after vascular injury by promoting VSMCs’ migration and inflammatory cell recruitment, thus serving as a potential therapeutic target for VSMCs’ migration and inflammation-related vascular diseases. Atox1 deficiency in ApoE^−/−^ mice on a Western diet showed a significant reduction in the formation of atherosclerotic lesions [104].

By using an unbiased system approach integrating sequencing data, genome-wide mapping of Atox1 regulatory targets in ECs was performed. Motif enrichment analysis and KEGG pathway enrichment analysis revealed CD137, colony-stimulating factor 1 (CSF1), and IL-5 receptor alpha (IL5RA) as new nuclear targeted inflammatory genes of Atox1, with CD137 also being a key regulator of ROS production induced by nucleus-targeted Atox1 (Atox1-NLS) [102]. These discoveries unveil novel downstream targets of nuclear Atox1 that play a role in inflammation and reactive oxygen species (ROS) generation, offering valuable insights into the potential of targeting nuclear Atox1 for treating inflammatory conditions like atherosclerosis. These discoveries highlight the importance of copper transport proteins in CVD and point out potential therapeutic pathways.

Copper-containing enzymes such as CP, SOD, and LOX play a role in mitigating the progression of AS through antioxidant effects or regulation of inflammatory responses.

Cp, also known as ferroxidase, is the major copper-carrying protein in the blood and is considered a potential biomarker for the risk of thrombosis in AS, with a close relationship between its levels and the development and progression of AS [105]. High levels of Cp may increase the risk of AS, while low levels could potentially reduce this risk. Specifically, Cp protects vascular endothelial cells from oxidative stress damage through its antioxidant properties, thereby slowing the progression of AS. Moreover, it can regulate inflammatory responses, and impact lipid metabolism and platelet aggregation, further affecting the formation of atherosclerotic plaques [105,133]. Therefore, maintaining appropriate levels of Cp is crucial for the prevention and treatment of AS.

Next, SOD is a metal-dependent antioxidant enzyme that eliminates superoxide radicals in the body by catalyzing the dismutation of superoxide into hydrogen peroxide and molecular oxygen, also showing a close relationship with AS [106]. In mouse models of atherosclerosis, SOD administration significantly reduced the formation and progression of atherosclerotic plaques. Additionally, SOD treatment improved lipid and glucose levels in mice, further alleviating the condition of AS [107].

Lastly, LOX is a copper-dependent enzyme that promotes the cross-linking of collagen and elastin in the extracellular matrix, which is vital for maintaining the structure and function of blood vessel walls [108,109]. Abnormal expression and activity of LOX in arterial endothelial cells have been associated with cardiovascular diseases, and LOX plays a role in regulating the proliferation of VSMCs and vascular remodeling [110,111]. However, when overactive, LOX promotes abnormal mineralization of the vascular wall, leading to atherosclerotic calcification through increased oxidative stress [112]. Overexpression and increased activity of LOX can result in excessive cross-linking and mineralization of the extracellular matrix, enhancing the stability of atherosclerotic plaques but also potentially increasing their brittleness and vulnerability, thus raising the risk of cardiovascular events [113,114]. Consequently, LOX has a dual role in the development of AS: it provides protective effects by enhancing plaque stability, but it also contributes to pathological processes by promoting plaque calcification and increasing their susceptibility to rupture. This discovery is significant for understanding the complexity of atherosclerosis and developing new therapeutic strategies.

Copper metabolism within the body is crucial for maintaining copper homeostasis, as elevated levels can induce proteotoxic stress, leading to cell death and significantly impacting the tricarboxylic acid cycle (TCA cycle) in mitochondria, thereby affecting energy production. As the primary energy source of the cell, mitochondria play a vital role in generating ATP through oxidative phosphorylation in the respiratory chain and fatty acid metabolism, supporting most cellular activities. However, excess copper can disrupt the biosynthesis and maturation process of iron–sulfur (Fe-S) cluster proteins in the mitochondrial respiratory chain, leading to impaired synthesis of the mitochondrial respiratory complex. Mitochondria are not only central to energy production but also a key target of copper-induced cell death. The damage inflicted by copper on mitochondria triggers oxidative stress and proteotoxic responses, which can influence endothelial injury, inflammatory reactions, proliferation or apoptosis of VSMCs, and macrophage polarization [134]. Concurrently, copper-induced cell death is closely associated with the formation and progression of AS. The mitochondrial damage and oxidative stress responses caused by excess copper may promote the development of CVD that severely affects the energy-demanding cardiovascular system [135,136].

PDHA1 and PDHB, along with DLAT and DLD, serve as key subunits of the pyruvate dehydrogenase complex (PDC) and play a significant role in the formation and development of AS [120,137,138,139]. PDHA1 and PDHB are components of the E1 subunit of the PDC, collectively catalyzing the conversion of pyruvate to acetyl-CoA. This process is not only a crucial link between glycolysis and the TCA cycle but also directly impacts fatty acid metabolism and energy production. Inhibition of PDHA1 expression or activity can disrupt intracellular energy balance and potentially accelerate the progression of atherosclerosis. Specifically, in endothelial cells, such inhibition can lead to an imbalance between glycolysis and oxidative phosphorylation, affecting cellular energy supply and function [140]. Shenmai injection can effectively protect myocardial mitochondria by enhancing the activity of pyruvate dehydrogenase and the antioxidant defense system. PDHA1, a key enzyme in aerobic metabolism, plays a significant role in the formation and development of AS [141]. Xuesaitong injection, by increasing PDHA1 activity, can promote aerobic metabolism and thereby reduce myocardial ischemia/reperfusion injury [122]. Meanwhile, changes in the activity of DLAT and DLD are also closely related to AS. DLAT, the E2 subunit of the PDC, is responsible for transferring the acetyl group to Coenzyme A to form acetyl-CoA, with its expression varying among individuals. DLAT activates the AMPK signaling axis in adipose tissue to produce heat, helping to suppress obesity [121]. Since obesity is a significant risk factor for AS, activation of DLAT can indirectly prevent AS. DLD, the E3 subunit of the PDC, acts as a core component of the complex, responsible for regenerating the oxidized form of lipoamide, namely converting dihydrolipoic acid (DHLA) back to lipoamide while transferring electrons to NAD^+^ to produce NADH [142]. DLD plays a crucial role in protein lipidation metabolism. Research has shown that deletion of the DLD gene can prevent lipid peroxidation and iron-dependent cell death. Inhibition of DLD expression increases intracellular ROS levels, leading to cell death. Additionally, DLD helps slow the development of AS by activating the RAS/ERK pathway and promoting the browning of white fat cells. Overexpression of DLD can also increase energy expenditure in white adipose tissue and improve dyslipidemia [119]. Furthermore, ginsenoside Rg3 exhibits protective effects against AS by regulating myocardial pyruvate metabolism [120]. In summary, the coordinated action of these subunits is crucial for maintaining mitochondrial energy metabolism balance and preventing AS.

FDX1, LIPT1, and LIAS collectively influence the development of AS by regulating mitochondrial function and lipid metabolism processes. FDX1 encodes an iron–sulfur protein that plays a crucial role in electron transfer within mitochondria. It interacts with ferredoxin reductase to transfer electrons from NADPH to mitochondrial cytochrome P450, which is vital for the synthesis and metabolism of steroids, vitamin D, and bile acids [143]. Moreover, FDX1 can reduce copper ions from a divalent to a monovalent state and participate in the biosynthesis of Fe-S clusters. In the process of protein lipidation, FDX1 acts as an upstream regulator of the LA pathway, controlling the post-transcriptional lipid modification of proteins such as PDC mediated by LIPT1 and LIAS, which is crucial for maintaining normal cellular activity and preventing copper-induced cell death [144]. FDX1 is a key CcO biosynthetic factor in mammalian cells and is highly expressed in atherosclerotic plaques. A newly discovered subset of macrophages also highly expresses FDX1, promoting inflammatory responses, affecting cholesterol metabolism, and contributing to the development of AS [115]. In patients with coronary artery disease, serum levels of FDX1 and LA are negatively correlated with the degree of coronary artery stenosis and the number of affected coronary artery branches, decreasing as the condition worsens. ApoE^−/−^ mice exhibit significantly elevated lipid levels and atherosclerosis indices, with noticeable thickening of the aorta, lipid accumulation, and collagen fiber proliferation, which are associated with reduced expression of FDX1, LA, LIAS, and aconitase 2 (ACO2) [116]. LIAS affects the occurrence and development of AS through various pathways, including regulating lipid metabolism, redox balance, and inflammatory responses. In the Lias^(+/−)^ApoE^(−/−)^males model, overexpression of LIAS significantly reduces the size of atherosclerotic lesions in the aortic sinus, increases the number of regulatory T cells and antioxidant LDL autoantibodies in the vascular system, and decreases T cell infiltration in the aortic wall [117,118]. Overexpression of LIAS also lowers plasma cholesterol levels, increases the activity of the PDC in the liver, and enhances the expression of antioxidant enzyme genes.

CDKN2A, GLS, and MTF1 have also been implicated in copper-induced cell death and can interfere with AS by regulating cellular metabolism, autophagy, and cellular senescence. Within atherosclerotic plaques, the expression of GLS is downregulated and primarily acts on smooth muscle cells. Through the regulation of yes-associated protein 1 (YAP1), the activity of GLS1 is inhibited, thereby helping to prevent ferroptosis in VSMCs [124]. Additionally, GLS1 can promote the clearance of dead cells by macrophages, exerting anti-inflammatory and antioxidant stress effects, maintaining cellular homeostasis, and thus slowing the progression of AS. MTF1, a transcription factor, can mitigate the risk of AS induced by a high-fat diet by modulating lipid metabolism. In studies on myocardial infarction due to coronary heart disease, MTF1 has been found to significantly impact the NF-κB signaling pathway, NOD-like receptor signaling pathway, and Toll-like receptor signaling pathway [96,123], which are closely related to inflammation and immune responses and play a crucial role in the occurrence and development of myocardial infarction in coronary heart disease. Recent studies have discovered that atherosclerosis-related cellular senescence offers new insights into the pathogenesis of AS [145]. The protein p16INK4a, encoded by the CDKN2A gene, plays a key role in the process of cellular senescence. CUT-like homeobox 1 (Cux1) regulates p16INK4a-dependent cellular senescence in ECs and VSMCs by binding to atherosclerosis-associated functional SNP (fSNP) rs1537371 on the CDKN2A/B gene locus [125,126]. Furthermore, in AS and type 2 diabetes, the mRNA levels of CDKN2A (p16Ink4a), CDKN2B (p15Ink4b), and CDKN2BAS are reduced [127]. Treatment with PD0332991, a p16Ink4a/p15Ink4b mimetic agent and a validated CDK4 inhibitor, enhances the expression of forkhead box protein 3 (FOXP3) and reduces lesion size in APOE^−/−^ mice and insulin-resistant ApoE^−/−^Irs2^+/−^ mice [128]. Therefore, CDKN2A, GLS, and MTF1 possess therapeutic potential for atherosclerosis.

## 6. Clinical Application of Copper Targeting Strategy in Atherosclerosis

In clinical practice, managing copper balance is crucial, as excessive or insufficient copper can have serious impacts on cardiovascular health. Identifying biomarkers that reflect changes in copper metabolism, such as copper content in plasma and serum, ceruloplasmin (Cp), and copper complex (SOD), can detect atherosclerosis risk early before clinical symptoms appear. This is crucial for timely intervention and improving patient prognosis. The changes in copper homeostasis in atherosclerotic diseases, combined with the discovery of copper death-related genes, are promising therapeutic targets for these diseases. In clinical applications, copper homeostasis is usually restored through two strategies: using copper homeostasis regulators to regulate copper bioavailability, and using copper homeostasis nanomedicines to maintain copper balance in the body.

### 6.1. Copper Homeostasis Regulators

Currently, copper homeostasis regulators used in the treatment of atherosclerosis (AS) primarily function by inhibiting vascular inflammation and improving mitochondrial function and energy metabolism. Tetrathiomolybdate (TTM), an orally active small-molecule anti-copper agent, forms complexes with copper and albumin in the blood, facilitating their excretion from the body. In ApoE^−/−^ mouse models treated with TTM, a significant reduction in aortic copper content was observed, effectively suppressing vascular inflammation and preventing the progression of AS lesions. Additionally, TTM inhibits TNF-α-induced activation of NF-κB and AP-1, reduces the mRNA and protein expression of VCAM-1, ICAM-1, and MCP-1, and diminishes endothelial cell activation [45,146]. Copper-aspirin, a commonly used analgesic and anti-inflammatory drug, exhibits enhanced anti-inflammatory effects when complexed with copper, showing strong therapeutic efficacy. As an effective antioxidant, it improves ventricular contractility in rat models of cardiovascular dysfunction by inhibiting oxidative stress and inflammatory signaling pathways [147]. Trientine (TETA), a highly selective copper chelator, has been shown to improve myocardial function in diabetic patients by restoring the activity of mitochondrial CCO, CCS, and SOD1 [148]. Moreover, inhibitors of the copper chaperones ATOX1 and CCS, such as DC-AC50, show potential in treating atherosclerosis (AS). DC-AC50 can block intracellular copper transport and, specifically, inhibit tumor cell proliferation without harming normal cells [149]. This action may slow down or prevent the formation of AS by reducing copper-mediated oxidative stress and inflammatory responses—key factors in the development of AS. Furthermore, ROS production induced by DC-AC50 may promote immunogenic cell death, trigger sustained immune responses, and affect mitochondrial function, providing new strategies for the treatment of AS.

### 6.2. Copper Homeostasis Nano-Regulators

The development and creation of nanomedicines designed to sustain copper homeostasis have demonstrated promising potential in the treatment of AS. Curcumin, a natural metal chelator, exhibits significant anti-inflammatory properties and can slow down the progression of AS through diverse signaling pathways. Recent studies suggest that targeted delivery techniques for curcumin hold promise for treating AS [17,150,151]. Polymer copper chelator RGD-PEG-b-PGA-g-(TETA-DTC-PHis) (RPTDH)/R848 nanoparticles effectively suppress the migration, proliferation, and angiogenesis of human umbilical vein endothelial cells by interrupting copper supply, thereby mitigating AS [152]. Dual-catalyst CuTPP/TiO2 nanoparticles exert their antithrombotic and regulatory effects on vascular wall cells (endothelial cells and smooth muscle cells), as well as their anti-inflammatory properties, by releasing nitric oxide (NO) signaling molecules and eliminating harmful reactive oxygen species (ROS), thus decelerating the progression of atherosclerosis [153].

## 7. Conclusions

This article provides an in-depth analysis of the intricate relationship between copper homeostasis and atherosclerosis, exploring how imbalances in copper homeostasis induce cell death that plays a role in atherosclerosis. The research indicates that disruptions in copper homeostasis not only jeopardize the integrity of the cardiovascular system but may also accelerate the progression of atherosclerosis by triggering inflammatory responses and oxidative stress processes. Additionally, a novel mechanism of cell death known as “cuproptosis” offers new insights into the role of copper in cardiovascular diseases. By studying genes related to copper homeostasis, we can uncover causal links between copper metabolism and cardiovascular diseases such as atherosclerosis on multiple levels, enhancing our comprehensive understanding of the mechanisms through which copper acts in the atherosclerotic process. Given copper’s crucial role in numerous physiological activities, developing safe and effective copper homeostasis regulators to maintain balanced copper levels in the body is an important trend for future research. Ultimately, with the continuous advancement of nanotechnology, employing nanomedicine delivery systems to precisely regulate copper homeostasis could emerge as an innovative strategy for treating cardiovascular diseases in the future.

## Figures and Tables

**Figure 1 ijms-25-11462-f001:**
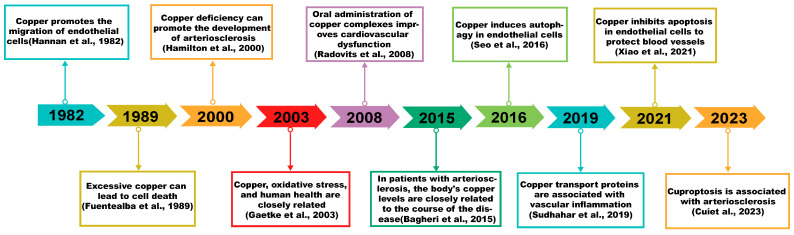
Copper and cardiovascular health. The timeline outlines significant historical milestones in the research advancements regarding copper’s association with AS [12,14,15,16,17,18,19,20,21,22].

**Figure 2 ijms-25-11462-f002:**
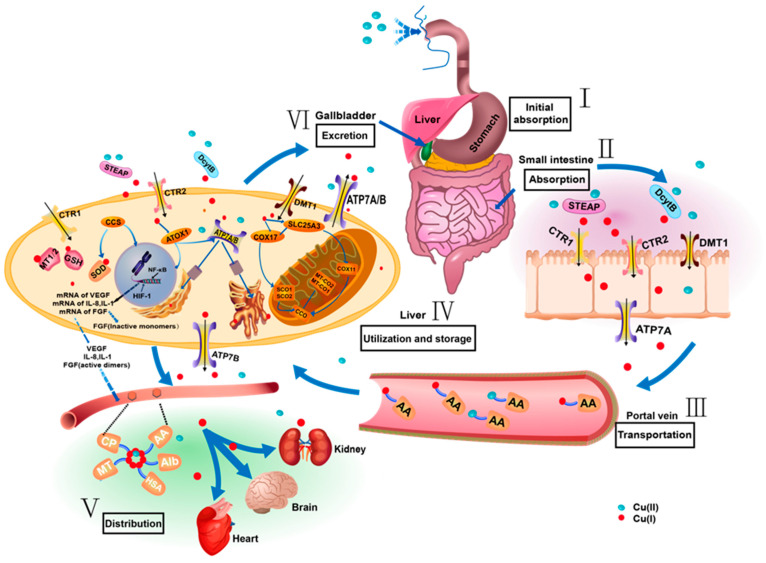
Metabolism and circulation of copper in the human body. This image meticulously outlines the metabolic process of copper in the human body, detailing every step from intake to excretion.

**Figure 3 ijms-25-11462-f003:**
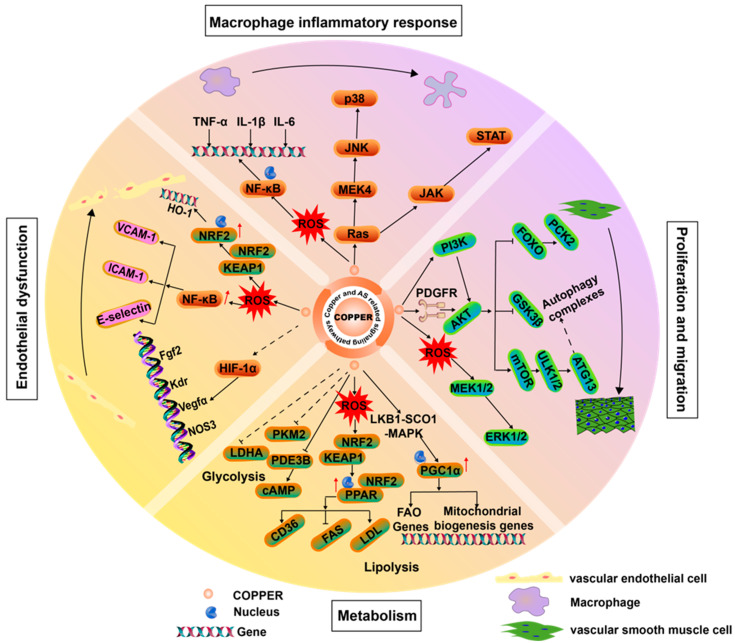
Copper and AS signaling pathways. Copper ions are critical in AS signaling, activating receptors and kinases like PDGFR, MAPK, and PI3K to promote smooth muscle cell proliferation and migration. They also trigger NF-κB and JNK/STAT pathways, releasing inflammatory cytokines TNF-α and IL-1β, which enhance macrophage inflammation and accelerate AS. Additionally, copper ions activate NRF2 through ROS and affect HIF-1α, regulating FGF2 and KDR genes linked to endothelial dysfunction. They can also modulate PDE3B, PKM2, and PPARγ, influencing cellular metabolism and AS formation.

**Figure 4 ijms-25-11462-f004:**
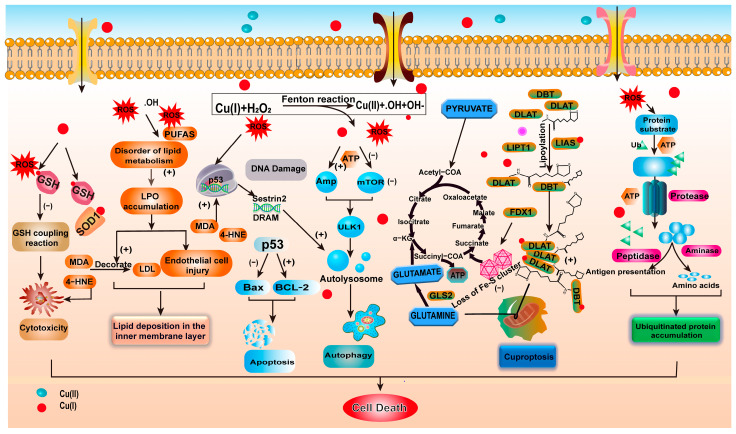
The mechanism of copper-induced cell death. The diagram elucidates the consecutive stages culminating in copper-induced cell death, beginning with the initiation of intracellular oxidative stress, advancing to DNA damage and disturbances in lipid metabolism, and finally leading to apoptosis and autophagy, in addition to a distinct mode of cell demise termed “cuproptosis”. This intricate process activates an array of signaling pathways and molecular mechanisms, highlighting the profound impact of copper ions on cellular processes.

## Data Availability

Data are contained within the article.

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
