# Peer review of "Deciphering the Role of Copper Homeostasis in Atherosclerosis: From Molecular Mechanisms to Therapeutic Targets"

_ijms, 2024, doi:10.3390/ijms252111462_

Round 1
Reviewer 1 Report
Comments and Suggestions for Authors
The present article is of interest and evaluate in deep the copper homeostasis in atherosclerosis.
Authors provide a really full description of the copper role in the pathogenesis of atherosclerotic diseases
The pictures provided are impressive and of easy access
My only suggestion is to include a small paragraph evaluating any potential clinical implication and how in the clinical practice we can manage copper haemostasis
Author Response
The present article is of interest and evaluate in deep the copper homeostasis in atherosclerosis.
Authors provide a really full description of the copper role in the pathogenesis of atherosclerotic diseases.
The pictures provided are impressive and of easy access.
Response: Thank you very much for your positive feedback and appreciation of our review. It is also gratifying to know that you found the provided pictures impressive and easily accessible.
My only suggestion is to include a small paragraph evaluating any potential clinical implication and how in the clinical practice we can manage copper haemostasis.
Response: We fully agree that discussing the potential clinical significance of copper balance and its importance in practical management strategies is crucial for enriching the content of the article and bridging the gap between basic scientific research and clinical practice. In response to your suggestions, we have added a dedicated paragraph to the article to evaluate the potential clinical significance and management strategies. Specifically, we have further emphasized the clinical value of relevant targets (such as SLC31A1, CP, SOD, etc.) in Table 1, exploring their potential as markers and therapeutic targets for atherosclerosis. Additionally, we have made significant revisions to Section 6, delving deeper into the significance of early detection biomarkers, therapeutic targets related to copper homeostasis, and strategies for restoring copper balance through regulators and nanomedicine.
Finally, we would like to express our sincere gratitude once again for your dedicated efforts in reviewing our manuscript.
Reviewer 2 Report
Comments and Suggestions for Authors
In the review, the authors described the physiological mechanisms of copper homeostasis and the role of it in the development of atherosclerosis. The article is well organized and has a logical structure. The figures and tables a re legible and add new information to the description given in the text. I congratulate the authors on the article and may recommend to accept the manuscript as it is.
Author Response
In the review, the authors described the physiological mechanisms of copper homeostasis and the role of it in the development of atherosclerosis. The article is well organized and has a logical structure. The figures and tables a relegible and add new information to the description given in the text. I congratulate the authors on the article and may recommend to accept the manuscript as it is.
Response: We sincerely appreciate your positive feedback and the thorough review of our manuscript. Although you have recommended accepting the manuscript in its current form, we will take this opportunity to carefully review the manuscript's logic and language during the revision process to further enhance its quality.
Finally, we would like to express our sincere gratitude once again for your dedicated efforts in reviewing our manuscript.
Reviewer 3 Report
Comments and Suggestions for Authors There are my answers: • The submitted manuscript entitled “Deciphering the role of copper homeostasis in atherosclerosis: from molecular mechanisms to therapeutic targets” reports on the role of copper, an essential trace element in the cardiovascular system. In this review article, the authors drew attention to copper homeostasis, copper-induced cytotoxicity, and its association with atherosclerosis.• The topic is original and relevant in monitoring the significance of essential elements copper in various process in cardiovascular system.
• The paper is well designed, follows the importance of copper in the states of homeostasis, with the attention on the metabolism and transport system of this essential element. On the other hand, the role of copper in pathological conditions is monitored. It is linked with the damage of endothelial cells, occurring due to activities of various kinase, which altogether leads to the damage of blood vessels.
• The authors did not specify the methodology. What data base did they used?
• The conclusion is in accordance with the results.
• The cited references are appropriate.
• The authors omitted “Figure 1. Copper and cardiovascular health”. I suggested that they incorporated missing figure in existing paper.
Author Response
There are my answers: • The submitted manuscript entitled “Deciphering the role of copper homeostasis in atherosclerosis: from molecular mechanisms to therapeutic targets” reports on the role of copper, an essential trace element in the cardiovascular system. In this review article, the authors drew attention to copper homeostasis, copper-induced cytotoxicity, and its association with atherosclerosis.
Response: We would like to express our sincere gratitude for your constructive comments and the time you dedicated to reviewing our manuscript.
- The topic is original and relevant in monitoring the significance of essential elements copper in various process in cardiovascular system.
Response: The positive feedback and constructive comments are highly appreciated.
- The paper is well designed, follows the importance of copper in the states of homeostasis, with the attention on the metabolism and transport system of this essential element. On the other hand, the role of copper in pathological conditions is monitored. It is linked with the damage of endothelial cells, occurring due to activities of various kinase, which altogether leads to the damage of blood vessels.
Response: Thank you for the positive feedback and constructive comments.
- The authors did not specify the methodology. What data base did they used?
Response: Thanks for your suggestion. We sincerely apologize for any confusion caused by not explicitly mentioning our research methodology. We conducted a comprehensive literature search using multiple databases including PubMed, Web of Science, and Google Scholar to collect data up to July 24, 2024. We have added this information to the "Introduction" sections to outline our review process and improve transparency.
- The conclusion is in accordance with the results.
Response: Thank you.
- The cited references are appropriate.
Response: Thank you.
- The authors omitted “Figure 1. Copper and cardiovascular health”. I suggested that they incorporated missing figure in existing paper.
Response: Thank you for pointing out the omission of Figure 1. We have included Figure 1 in the "Introduction" section.
Finally, we would like to express our sincere gratitude once again for your dedicated efforts in reviewing our manuscript.